# Motivational effectiveness of prosocial public health messaging to reduce respiratory infection risk: a systematic review and meta-analysis
Aikaterini Grimani [1] ✉, Vivi Antonopoulou[2], Nicholas Meader[3], Chris Bonell[4], Susan Michie [2], Michael P. Kelly [5,6] & Ivo Vlaev[7]

## Abstract

**Background** Clear communication is essential for the effective uptake of public health interventions promoting protective behaviours for respiratory infection control. The emergence of novel infectious diseases, particularly the COVID-19 pandemic, has highlighted the need for rapid adaptation of established and new behavioural practices. However, there remains limited knowledge concerning effective strategies for disseminating risk-reduction information and predicting population responses.

**Methods** This systematic review and meta-analysis (PROSPERO: CRD42020198874) assessed the effectiveness of these interventions using behavioural science frameworks, including MINDSPACE contextual influencers and behaviour change techniques (BCTs), to identify key components and mechanisms of action (MoAs). Twenty-four full-text articles, comprising 36 randomised controlled trials (RCTs) across 11 countries, were included via electronic databases (MEDLINE, EMBASE, PsycINFO, Scopus) and other sources (grey literature, Google Scholar, and reference lists) searched to March 2022.

**Results** Here, we show that interventions mainly target social distancing, mask wearing, hand washing, and various behavioural intentions and actual behaviours, using a median of three-arm study designs with passive comparators. Interventions include a median of two contextual influencers and four BCTs. Behaviour intention is the most frequently applied mechanism of action. Study quality is moderate. Narrative synthesis of 16 full-texts (26 RCTs) shows significant effects, while network meta-analysis of 16 full-texts (21 RCTs) indicates that prosocial messages, particularly those referencing loved ones, are effective in reducing the risk of respiratory infections (d = 0.09; 95% CrI=0.06–0.14; CINeMA: Low).

**Conclusions** Although further research is needed, the review provides insight into designing public health messages that effectively improve protective behaviours for respiratory infection control.

## Plain language summary

This study examined whether public health messages can encourage people to adopt protective behaviours, such as wearing masks, washing hands, and keeping distance, to reduce the spread of respiratory infections like COVID-19 and influenza. We reviewed and combined the results of 36 studies from 11 countries. To understand what works best, we used behavioural science tools that show how messages influence people's decisions and actions. We found that messages appealing to social values, especially those about protecting loved ones, were more effective in encouraging safer behaviours. These findings highlight the importance of designing health messages that connect with people's emotions and motivations. Clear and well-targeted communication can help the public respond more effectively in future outbreaks.

The COVID-19 pandemic made plain that in the face of a novel viral infection causing severe respiratory disease, among other containment measures, the rapid adoption of population protective and physical social distancing behaviours was required[1,2]. As the waves of infection unfolded and vaccinations became available, it remained the case that behavioural interventions focused on personal protective behaviours (such as hand and respiratory hygiene, mask wearing, and social distancing) were still important to contain transmission[3–5]. Respiratory infection is, of course, not just the consequence of COVID, and into the future, containment strategies against influenza in particular will continue to be important.

Clear communication is essential for the development of public health advice messages that promote effective behavioural respiratory infection

control. Understanding what influences the uptake of non-pharmaceutical interventions may help to inform the development of future effective public health advice messages[1,6]. For instance, simply advising people to adopt these behaviours has been found to be ineffective[7]. Consequently, in the context of both pandemics and seasonal epidemics, it becomes imperative to employ effective strategies for conveying vital health messages in a succinct and meaningful manner. These strategies should facilitate the adoption of protective behaviours by citizens while mitigating community ambivalence and panic[8,9].

A promising key principle, which is based on social identity[10], social influence[11] and moral behaviour[12], promotes care for others rather than individual self-interest[13]. These 'protect each other' messages highlight the benefits of protective behaviours for the wider social group and its most vulnerable members, including loved ones, with evidence of benefits in the COVID-19[14] and other health contexts[15]. Prosocial behaviour is defined as behaviour that benefits others, whether or not it involves an overall cost to self, and includes a variety of important social behaviours such as helping, sharing, and cooperation[16,17]. The term "others" refers to specific individuals or groups of people, and in particular loved ones, vulnerable members (with weakened immune systems such as the elderly and chronically ill), health care professionals, co-workers, keyworkers, members of the public, and to some extent society as a whole[18].

Studies have shown that people do consider the social impact of their behaviour during a pandemic, as many participants avoid putting others at risk for their personal benefit[19]. The existing literature illustrates that pro-social public health messages that highlight behaviours related to societal and communal benefits (e.g., "protect each other"), rather than focusing on behaviours that only benefit the self (e.g., "protect yourself"), can be a potential mechanism to communicate public health recommendations related to infectious diseases[20–23].

Although the studies mentioned earlier provide some evidence for the effectiveness of prosocial messages in promoting adherence to personal protective behaviours, we still lack robust evidence about the magnitude of this effect[24]. As individuals are exposed to con-flicting information and misinformation, this can decrease individuals' perceptions of the severity of the disease and the necessity of adopting preventive behaviours, which in turn may lead to rejection of public health messaging. Therefore, there is a need to examine the extent to which prosocial messages to adhere to personal protective behaviours are effective across different contexts, for better future pandemic and epidemic preparedness. We conducted a systematic review and meta-analysis to identify whether the prosocial messages have the potential to optimise the effect on population behaviour in relation to reducing transmission of respiratory infections. More specifically, we ask the following questions: (1) Are messages focusing on protecting others effective in changing a defined list of beha-vioural (intention) outcomes compared with other messages/con-trols?, (2) What actual behaviours or behaviour intentions (e.g., social distancing, hand washing, using hygiene products etc.) do messages about protecting others have positive effects on? (3) What popula-tions do messages about protecting others have positive effects on?

A total of 36 randomized controlled trials conducted in 11 countries met the inclusion criteria for the systematic review (21 were included in the meta-analysis). The network meta-analysis shows that prosocial messages have a small but statistically significant effect on personal protective beha-vioural intentions (d = 0.09, 95% CrI 0.06–0.14). Self-focused messages show weaker and less consistent effects. Component network meta-analyses indicate that interventions incorporating information about health conse-quences (d = 0.12, 95% CrI 0.03–0.28) and avoidance or reduction of exposure to cues (d = 0.19, 95% CrI 0.06–0.32) are associated with small but statistically significant increases in protective behavioural intentions. Con-sidering these results, prosocial and carefully designed communication strategies should be prioritized in future public health campaigns to strengthen behavioural intentions for respiratory infection control.

## Methods

This systematic review was registered with PROSPERO (CRD42020198874)[25]. A detailed protocol of the review has been published[18]. We used established PRISMA (Preferred Reporting Items for Systematic Reviews and Meta-analyses) guidelines statement for Network Meta-Analyses[26] and also followed quantitative meta-analysis reporting standards by the APA[27] and procedures outlined by the Cochrane Hand-book for Systematic Reviews[28]. Institutional Review Board (IRB) approval was not required for this systematic review, as the study utilized exclusively previously published literature and did not involve the collection of primary data.

### Search strategy and selection criteria

A search strategy (Supplementary Table S1) following PICOS was adapted (population, intervention, comparison, outcome, and study design), including Medical Subject Headings (MeSH) terms and relevant keywords and combining search terms using Boolean operators[29,30]. To identify con-trolled vocabulary terms for the databases, we first retrieved articles from each database that met the inclusion criteria for the review. We noted common text words and subject terms applied by the indexers and used them for a comprehensive search. To ensure we captured as many relevant records as possible, including older ones, we combined subject terms from the controlled vocabulary with a wide range of free-text terms[31].

Original studies were eligible for inclusion if they met the following criteria: (a) were RCTs, non-randomised studies with concurrent controls, or controlled before-and-after studies; (b) measured potential changes in any behaviour relevant to reducing the transmission of respiratory infec-tions (e.g., hand hygiene, social distancing, face masks); (c) included mes-sages with prosocial content about protecting others[17]; (d) used communication via mass media, social media, or print media (such as leaflets and posters), or health professional advice via consultation; (e) included general population from any geographical region, with or without vulnerabilities, regardless of infection status; (f) included relevant compar-isons involving no message, or an active control with messages focused on self-protection, or messages containing no motivational content and (g) were published in English (Supplementary Table S2). Studies such as animal experiments, abstracts, case reports, reviews, and systematic reviews were excluded.

A comprehensive literature search of published and unpublished stu-dies in electronic databases was conducted. The following electronic data-bases were searched from inception to January 2021 and updated in March 2022: Medline, EMBASE.com, PsycINFO, and Scopus. For unpublished studies, we conducted searches in databases of "grey" literature such as PsycEXTRA, Social Science Research Network (SSRN), OSF PREPRINTS database that includes BioHackrXiv, Cogprints, MediArXiv, SocArXiv, PsyArXiv and RePEc. We also conducted supplementary searches in Google Scholar, hand searched relevant journals, and backward and forward cita-tion searching of included studies and relevant reviews. For each grey lit-erature source, we developed tailored search strategies, adapted from those used in electronic databases, using combinations of controlled vocabulary (where available) and free-text terms aligned with our PICOS framework, combined with Boolean operators. For Google Scholar, which uses pro-prietary ranking algorithms and can retrieve extremely large numbers of results even with highly specific search criteria, we limited screening to the first 10 pages (100 results) sorted by relevance. We applied the same approach to OSF Preprints (including BioHackrXiv, Cogprints, MediArXiv, SocArXiv, PsyArXiv, and RePEc), which also return very large numbers of records similar to Google Scholar. This decision was guided by both methodological precedent and practical considerations. Empirical studies indicate that the most relevant and highest-quality results are concentrated within the first 100–200 hits, with subsequent pages yielding progressively less relevant material and a higher proportion of false positives[32,33]. Screening beyond this point is unlikely to substantially improve compre-hensiveness but would considerably increase workload and introduce

irrelevant results. Our approach is consistent with established guidance for systematic reviews using search engines, where restricting screening to top-ranked results is recognised as a practical and defensible strategy. For backward citation searching, we manually reviewed the reference lists of all studies sought for retrieval as well as of relevant systematic reviews to identify additional potentially eligible records. For forward citation searching, we used Google Scholar to identify more recent studies that had cited these articles.

All titles and abstracts retrieved from both published and unpublished studies through electronic searching were imported into the reference manager EndNote and subsequently uploaded to Covidence, a systematic review management tool recommended by Cochrane, to facilitate duplicate removal, screening, and study selection[34]. Duplicates were removed and double-screening was done on a proportion of the retrieved citations until the appropriate agreement (>95%) was achieved (450 studies; kappa = 0.969; 95% Cl 0.909–1.000). The remaining studies were screened by one reviewer (AG). The abstracts were included if they met the inclusion criteria. The same procedure was applied to determine the eligibility of studies on the basis of a review of the full texts. Differences in judgement were resolved through discussion and inclusion of a third researcher doing the rating, when required. The selection process was recorded and the PRISMA flow diagram was completed[35].

### Data extraction
Data were extracted from the included studies into a predefined spreadsheet, which covered the following areas: the study design; the communication message; characteristics of the recipient(s) of the communication (protect-others-message); characteristics of the "others" who will be protected due to the message; the manner in which the appeal to protect others is made; intervention features such as the primary outcome(s) and the results.

As primary outcomes, we considered any actual behaviour or behavioural intention relevant to reducing transmission of respiratory infections. Intentions are assumed to capture the motivational factors that influence behaviour. The stronger the intention to perform the behaviour, the more likely the behaviour will be performed[36,37]. Where more than one reported primary outcome is provided, we included all, reporting their measurements, metrics, methods of aggregations and time-points (when applicable).

Study characteristics were extracted by one reviewer (AG) and checked for accuracy by a second reviewer (VA). Outcome data were extracted independently by two reviewers (AG and VA). Any discrepancies were resolved through discussion with a third reviewer (IV).

### Quality assessment
Two researchers (AG and VA) independently assessed the risk of bias of the included studies using the Cochrane Collaboration Risk of Bias Tool (RoB 2)[38]. The percentage of studies with high, low, or some concerns against each criterion was established by consensus between the two assessors. No studies were excluded because of risk of bias.

### Data analysis and synthesis
We carried out a systematic review to qualitatively extract the key MINDSPACE conceptual influencers and behaviour change techniques used in prosocial messaging interventions to adhere to personal protective behaviours and quantitatively assess their effectiveness in reducing transmission of respiratory infections.

A narrative synthesis was conducted for each of the personal protective behavioural intentions and actual behaviours. Reviewers' own descriptions of results were grouped into "positive" effects (e.g., a significant difference between intervention and control groups, favouring the intervention group, was detected), "no difference" (e.g., a significant difference between groups was not detected), "negative" effects (e.g., a significant difference between groups, favouring the control group, was detected). A narratively synthesised section with the attributes of the target population (characteristics of the individuals, groups, sub-populations or populations) that are affected by messages about protecting others was also included.

We content analysed these messages to identify what drives the change in behaviour. Two of the most popular and widely used behavioural tools were chosen due to their relevance and applicability to health policy, the MINDSPACE checklist[39–41] (stands for: Messenger, Incentives, Norms, Defaults, Salience, Priming, Affect, Commitments, and Ego; Supplementary Table S3 and Supplementary Note 1), and the Behaviour Change Techniques Taxonomy (BCTT; Supplementary Fig. 1 and Supplementary Note 2)[42–46]. The MINDSPACE checklist, developed by Dolan et al.[39], operates on the premise that many behaviours are primarily driven by automatic, often unconscious psychological processes and can be influenced by the decision-making context[41]. The BCTT includes 93 techniques aimed at fostering behaviour change by targeting specific psychological processes, including beliefs, attitudes, goals, plans, emotions, habits, and more. There has not been an academic investigation of the degree of conceptual (or theoretical) overlap between MINDSPACE and the BCTT (see Vlaev and Dolan[47] for initial analysis of how those nudges and techniques work through specific, and distinct, neuropsychological processes). Therefore, the possibility for unique insights to exist in each behavioural tool motivated us using both tools in our analysis. To address the challenges of inconsistent or ambiguous terminology in behaviour change intervention outcome measures, we utilized the "Mechanisms of Action" (MoA) Ontology[48,49]. This tool acts as a classification system that labels and defines MoAs and their interrelationships. MoAs describe the processes by which interventions influence target behaviours, such as beliefs, intentions, and behavioural opportunities (Supplementary Method; Supplementary Data 1).

We also calculated the effective ratio (ER) to assess the effect of each MINDSPACE contextual influencer and BCT for the USA and European countries. This measure indicates their potential contribution to the effectiveness of the intervention[50]. The intervention outcomes were classified as effective (indicating statistical significance in intervention(s) and control comparisons) and ineffective (outcome was not statistically significant). The ER is calculated as the number of effective results involving the MINDSPACE contextual influencer or BCT (i.e., interventions that were statistically significantly more effective than the control) divided by the number of ineffective results involving that same influencer or technique. The ER could not be calculated if only one study was available.

A network meta-analysis was conducted as most appropriate for comparing three or more interventions simultaneously in a single analysis by combining both direct and indirect evidence across a network of studies, calculating the standardized mean difference (SMD) with 95% credible intervals (CrIs: the Bayesian alternative to 95% confidence intervals) using WinBUGS, a statistical software for Bayesian analysis. All analyses included 30,000 iterations as burn-in; once convergence was confirmed, these iterations were discarded and the model was run for a further 70,000 iterations.

It produces estimates of the relative effects between any pair of interventions in the network, and usually yields more precise estimates than a single direct or indirect estimate[51]. The simultaneous comparison of all interventions of interest in the same analysis enables the estimation of their relative ranking for a given outcome[52]. In particular, a class-effects model[53] was conducted, including all the outcomes reported in RCTs (which met the eligibility criteria for inclusion in the meta-analysis: adequate result information, consistency in outcome elements (e.g., consistent methods of aggregation, consistent specific measurements and metrics)) whilst accounting for dependencies where more than one outcome per RCT is included. In addition, random effects NMAs were calculated using WinBUGS for each outcome separately. A key NMA assumption is the consistency between direct and indirect evidence. Global tests for inconsistency were conducted for each outcome using a design x treatment interaction test[54]. When a global test identified potential inconsistency, tests for loop inconsistency were conducted for each evidence loop. Network geometry was explored and illustrated using network diagrams. Component network meta-analyses (CNMAs) were conducted to assess the effectiveness of intervention components[55,56] categorised according to the BCTT and Mindspace components. Class effects were also used to account for multiple outcomes reported in RCTs.

For each outcome, including personal protective behavioural intentions and subgroup outcomes like social distancing intentions, hand washing intentions, mask wearing intentions, and diverse-behavioural intentions, we evaluated the confidence in the body of evidence derived from NMA using the Confidence in Network Meta-Analysis (CINeMA) application[57,58], which is broadly based on the Grading of Recommendations Assessment, Development, and Evaluation (GRADE) approach. The GRADE methodology considers four levels of evidence: high quality, moderate quality, low quality, and very low quality. High quality indicates that the true effect closely aligns with the NMA estimates, while moderate quality suggests a likelihood that the actual effect is similar to the NMA estimates but could differ substantially. Low quality implies a possibility that the true effect differs significantly from the NMA estimates, and very low quality indicates a high likelihood of substantial differences. We determined the certainty of the evidence using the online CINeMA software (https://cinema.ispm.ch), which assesses criteria such as within-study bias, reporting bias, indirectness, imprecision, heterogeneity, and incoherence. If any concerns, whether minor or major, are identified, the certainty of the evidence is downgraded. Although CINeMA is a valuable tool for assessing the certainty of evidence in NMAs, its direct applicability to CNMAs may be limited due to differences in analytical approach and scope of assessment. Thus, the CINeMA has not been used for our CNMAs.

## Results

### Study selection and characteristics
The search strategy yielded 6108 studies. After screening the titles and abstracts, 49 studies were selected for full-text screening. A total of 24 full-texts met the inclusion criteria (20 were published[16,24,59–76], while 4 were pre-prints)[14,77–79]. Eight of them included more than one randomised controlled trial study (RCT), yielding 36 RCTs in total. No studies with other designs (such as quasi-experimental studies) fulfilled the inclusion criteria. The majority of RCTs (32/36) targeted COVID-19, while 4 RCTs targeted multiple respiratory infections, including (but not limited to) influenza. The characteristics of included RCTs are described in more detail in Supplementary Data 2 and 3. Sixteen (21 RCTs)Study[1,14,24,62,63,65–72,74,76,78,79] of the 24 full-texts (36 RCTs) provided sufficient information to be included in the final quantitative analysis (Fig. 1).

### Effectiveness of prosocial messages
Twenty-six RCTs from high-income countries (USA, Denmark, UK, Japan, Germany, Turkey) reported positive effects regarding the included messages focusing on "protect-others" principle[16,24,60]Study[1,61–63,66,68]Study[1,69–73,75,76,78], one RCT (from USA) reported negative effects[59] and nine RCTs (from USA, Italy, France and multiple countries: Spain, Chile, Colombia) reported no differenceStudy[3,14,60,64,65,67,68,74,77,79]. None were in low or middle-income countries. Seventeen of the RCTs reporting positive effects regarding the included messages focusing on the "protect-others" principle were conducted in the USA, five in Europe, and four in Asia (see Supplementary Results 1). The main outcomes that positively affected by the messages focusing on protecting others were social distancing intentions (e.g., motivation to adhere to physical distancing, persuasiveness to self-isolate, avoid social gathering, stay home and keep a physical distance with others), mask wearing intentions, hand washing intentions, diverse-behavioural intentions (intentions that could not be categorized into specific groups, such as contact-avoidance intentions, protective behaviour willingness) and actual behaviours (see Supplementary Results 2).

### Behavioural Tools and Mechanisms of Action (MoA) ontology evidence
Six of the nine MINDSPACE contextual influencers were identified across 102 intervention arms. On average, each intervention arm adopted 2.52 MINDSPACE contextual influencers. The four most common MIND-SPACE contextual influencers were "Messenger" ($n = 33$; 32%), "Salience" ($n = 96$; 94%), "Affect" ($n = 42$; 41%), "Ego" ($n = 43$; 42%) (Supplementary

Table S4; Supplementary Note 3). The most often applied contextual influencers across the 74 intervention groups focused on protecting others were "Salience" ($n = 53$; 72%) and "Ego" ($n = 32$; 43%). "Affect" was also present, however, less frequently ($n = 20$; 27%).

Twenty-eight of the 93 BCTs were identified across 102 intervention arms. On average, each intervention arm adopted 4.03 behaviour change techniques. The seven most common BCTs identified were "Instruction on how to perform a behaviour" ($n = 96$; 94%), "Information about health consequences" ($n = 84$; 82%), "Salience of consequences" ($n = 69$; 68%), "Information about social and environmental consequences" ($n = 28$; 27%), "Credible source" ($n = 34$; 33%), "Avoidance/reducing exposure to cues for the behaviour" ($n = 26$; 26%) and "Prompts/cues" ($n = 18$; 18%) (Supplementary Table S5; Supplementary Note 4). Twenty-five of the 28 BCTs identified across the 74 intervention arms focused on protecting others, with the following as the most frequently applied: "Instruction on how to perform a behaviour" ($n = 54$; 73%), "Information about health consequences" ($n = 51$; 69%), "Salience of consequences" ($n = 44$; 60%), "Avoidance/reducing exposure to cues for the behaviour" ($n = 19$; 26%), "Information about social and environmental consequences" ($n = 18$; 24%) and "Credible source" ($n = 14$; 19%).

MoA Ontology applied to 21 of the 24 full-text papers, covering 30 MoA subcategories. On average, each study adopted 3.48 MoAs, with "Behavioural intention" being the most common, followed by "Belief about one's social environment" and "Mental disposition" (Supplementary Table S6; Supplementary Note 5; Supplementary Table S7).

### Cross-country analysis
Due to the lack of data and the inability to perform a meta-analysis to assess the impact of each MINDSPACE contextual influencer and BCT across the USA and European countries, we calculated the Effective Ratio (ER) to strengthen our narrative analysis. In RCTs conducted in the USA, we evaluate the impact of five MINDSPACE contextual influencers (Fig. 2) and nine BCTs (Fig. 3). The most common MoAs were "Behavioural intention" and "Belief about message" followed by "Belief about consequences of behaviour" and "Willingness to comply". In RCTs conducted in European countries, we evaluate the impact of four MINDSPACE contextual influencers (Fig. 2) and four BCTs (Fig. 3). The most common MoA was "Behavioural intention", followed by "Motivation", "Emotion process", "Evaluative belief about behaviour" and "Belief about control over behaviour". In RCTs conducted in Japan and Turkey, ER values could not be estimated for either MINDSPACE contextual influencers or BCTs, as there were no ineffective results.

### Populations' characteristics affected by prosocial messages
Demographic characteristics appear to influence how individuals respond to prosocial public health messages encouraging protective behaviours. While seventeen studies either found no significant effects or did not include demographic analysis, six studies identified key predictors of responsiveness, namely age, gender, employment status, political orientation, race, education, and health condition. In particular, older adults, women, individuals in poorer health, more religious individuals, and those with liberal political views were more responsive, showing greater intention to adhere to protective behaviours.

Age emerged as a consistent predictor across multiple studies. Older individuals were more likely to report intentions to engage in protective behaviours such as handwashing, mask-wearing, and social distancing. Browning et al.[78] and Capraro and Barcelo[62] both found that older age was associated with stronger intentions to follow COVID-19 prevention guidelines (e.g., wearing face coverings). Similarly, Everett et al.[14] and Hacquin et al.[79] found that older adults were more inclined to adopt behaviours like handwashing and distancing compared to younger individuals. These findings suggest that older populations are more susceptible to prosocial messaging aimed at reducing transmission, possibly due to higher perceived risk and community responsibility. No studies in our review reported older age as a negative predictor.

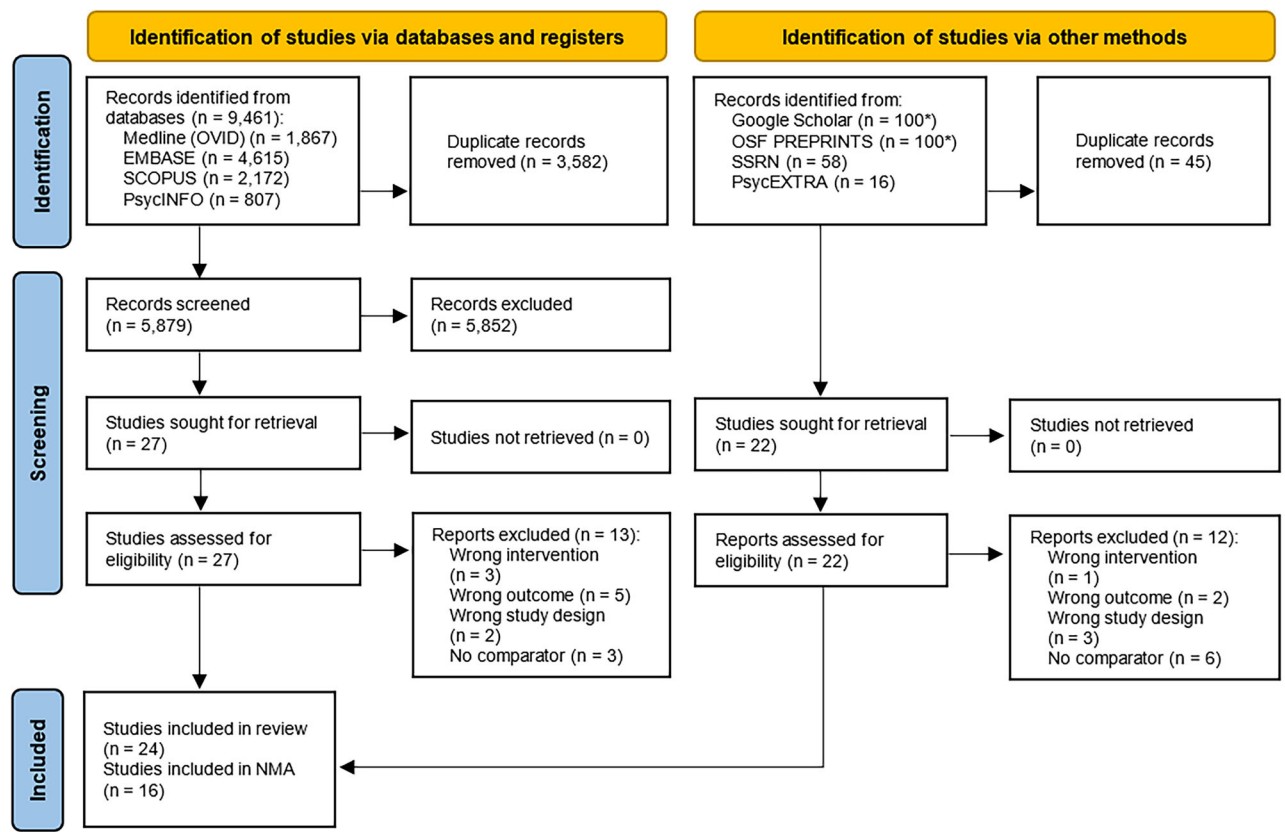

**Fig. 1 | PRISMA Flow Diagram[93].** * First 100 references from each database and search engine have been kept sorted by relevance.

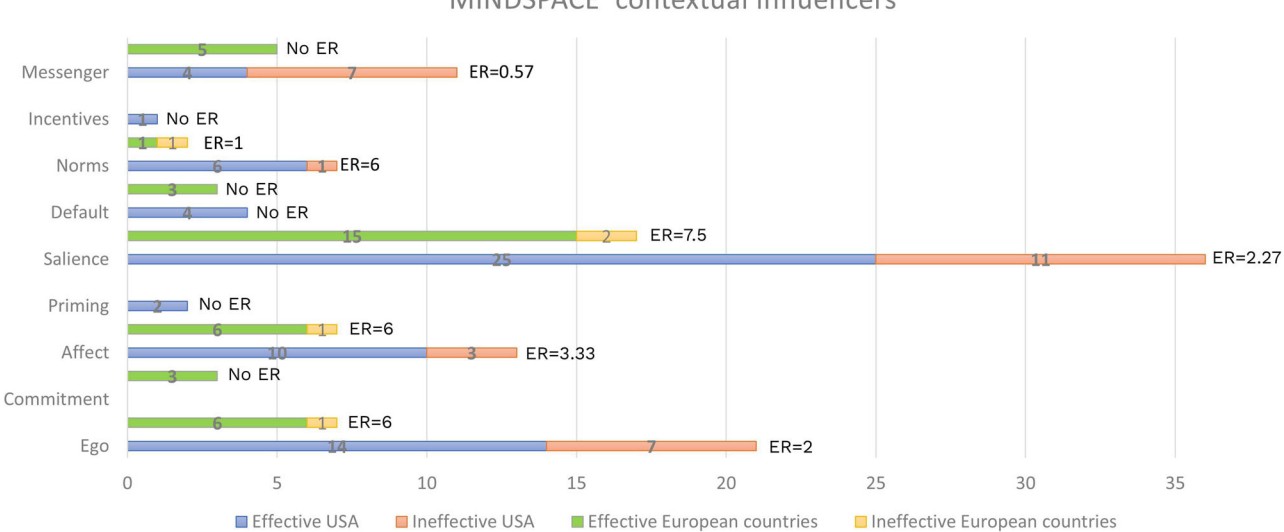

**Fig. 2 | ERs of MINDSPACE contextual influencers; ER = effective ratio.**
**USA**: The highest ER was observed for interventions focusing on "Norms" (ER = 6), followed by "Affect" (ER = 3.33), "Salience" (ER = 2.27), "Ego" (ER = 2) and "Messenger" (ER = 0.57). For four MINDSPACE contextual influencers, ER values could not be estimated due to the absence of ineffective results. **European countries**: The highest ER was observed for interventions focusing on "Salience" (ER = 7.5), followed by "Affect" (ER = 6), "Ego" (ER = 6) and "Norms" (ER = 1). For five MINDSPACE contextual influencers, ER values could not be estimated due to the absence of ineffective results.

Gender also played a role, with women generally reporting stronger behavioural intentions and greater responsiveness to messaging compared to men. Pink et al.[60] found that women increased their intention to comply after viewing prosocial messages more than men. Hacquin et al.[79] and Everett et al.[14] similarly identified women as more likely to engage in handwashing and other protective behaviours. However, Capraro and

Barcelo[62] found that this gender difference reduced when mask-wearing became mandatory, suggesting external regulations may override gender-based behavioural patterns.

Health status influenced message effectiveness, particularly among individuals in poorer health or at higher risk of infection. Falco and Zaccagni[61] found that these individuals were more likely to be influenced by

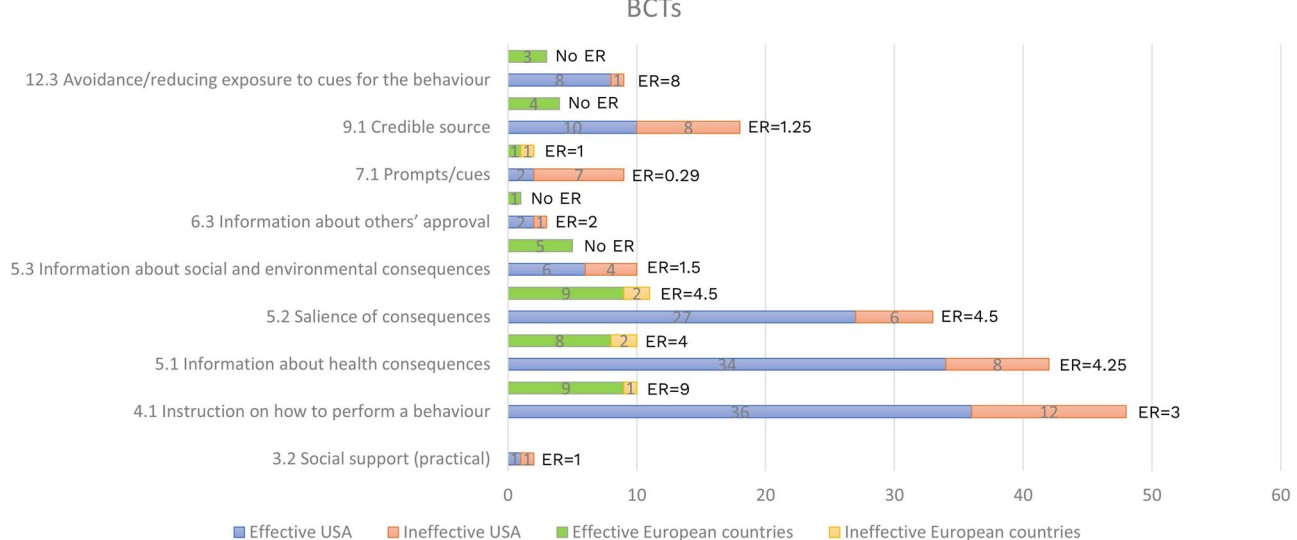

**Fig. 3 | ERs of behaviour change techniques (within letters); ER=effective ratio.**
**USA**: The highest ER in interventions pertained to "Avoidance/reducing exposure to cues for the behaviour" (ER = 8), followed by "Salience of consequences" (ER = 4.5), "Information about health consequences" (ER = 4.25), "Instruction on how to perform a behaviour" (ER = 3), "Information about others' approval" (ER = 2), "Information about social and environmental consequences" (ER = 1.5), "Credible source" (ER = 1.25), "Social support (practical)" (ER = 1) and "Prompts/cues"

(ER = 0.29). For six BCTs, ER values could not be estimated due to the absence of effective results. **European countries**: The highest ER in interventions pertained to "Instruction on how to perform a behaviour" (ER = 9), followed by "Salience of consequences" (ER = 4.5), "Information about health consequences" (ER = 4) and "Prompts/cues" (ER = 1). For 19 BCTs, ER values could not be estimated due to the absence of effective results.

messages emphasizing the protection of others, such as family members. Conversely, those in good health or who frequently left their homes were less impacted by such messages.

Political orientation was a consistent factor influencing behavioural intentions. Individuals who identified as more liberal or left-leaning tended to show greater behavioural intentions following exposure to prosocial messages (Capraro and Barcelo[62]; Pink et al.[60]). In contrast, those who identified as conservative or right-leaning were generally less responsive, reporting lower behavioural intentions and a reduced sense of personal responsibility to prevent the spread of disease (Everett et al.[14]).

Religiosity was also associated with behavioural intentions. Religious individuals reported stronger intentions to engage in protective behaviours. According to Everett et al.[14], religiosity was linked to greater perceived responsibility and motivation to follow public health guidance, suggesting that religious values may align with collective-oriented health behaviours.

Employment status, though less frequently studied, appeared to play a role in behavioural intentions. Browning et al.[78] found that individuals in less secure employment were more likely to report higher intentions to engage in COVID-19 preventive behaviours. This suggests that perceived vulnerability or instability may heighten sensitivity to public health messaging.

The role of education in behavioural intentions was more complex. Hacquin et al.[79] reported that individuals with lower education levels tended to show greater responsiveness to prosocial messaging and higher intentions to follow protective behaviours such as handwashing. However, in some instances, higher education levels were associated with lower behavioural intentions, indicating a nuanced relationship.

Ethnicity also seems to be a recurring factor. Browning et al.[78] reported that individuals identifying as White or White/Indigenous demonstrated lower intentions to engage in preventive behaviours. Similarly, Everett et al.[14] found that White-identifying individuals reported weaker behavioural intentions and felt less personally responsible for preventing disease transmission.

### Risk of bias
Agreement between the two independent raters in coding the risk of bias criteria was high (91,7%). Overall, we noted a high level of some

concerns, which was the result of insufficient reporting of an appropriate analysis used to estimate the effect of assignment to the intervention. An intention-to-treat (ITT) analysis that includes all randomised participants was lacking, resulting in some concerns in all studies. Further, some of the studies did not publish a protocol or register the studies on trial registries, making it difficult to assess reporting bias, and this led to downgrading of the evidence quality for the large majority of the included intervention types. The studies were assessed as low risk of bias for the domains "bias arising from the randomisation process", "bias due to missing outcome data" and "bias in measurement of the outcome". Only one study assessed as high risk of bias due to missing outcome data (Fig. 4; Supplementary Data 4).

### Network meta-analysis
Twenty-one RCTs with behavioural intention outcomes (social distancing, mask wearing, handwashing and diverse-behavioural intentions) were considered for quantitative analyses. None of the studies with actual behaviour outcomes met the eligibility criteria for inclusion in the meta-analysis. Nineteen RCTs ($n = 30$ intervention arms (prosocial messages focused on public and loved ones; self-focused messages); $n = 19$ comparator groups) were included in the class-effects model, including multiple behavioural intention outcomes except for the handwashing intentions outcome, due to inconsistency between direct and indirect evidence. There was a small increase in personal protective behavioural intentions for each intervention group (prosocial messages and self-focused messages) compared to the control group (no message, baseline message or self-protection message), based on effect estimates of standardised mean differences (SMDs) with 95% CrI. Although the effects of all three intervention messages were small compared to the control and closely comparable to each other, prosocial messages focused on loved ones showed a slightly higher effect size ($d = 0.09$, 95% CrI 0.06 to 0.14, CINeMA: Low). Additionally, the 95% CrI for the comparison between prosocial messages focused on loved ones versus control did not cross 0, indicating that a positive association between prosocial messages focused on loved ones compared to the control group exists

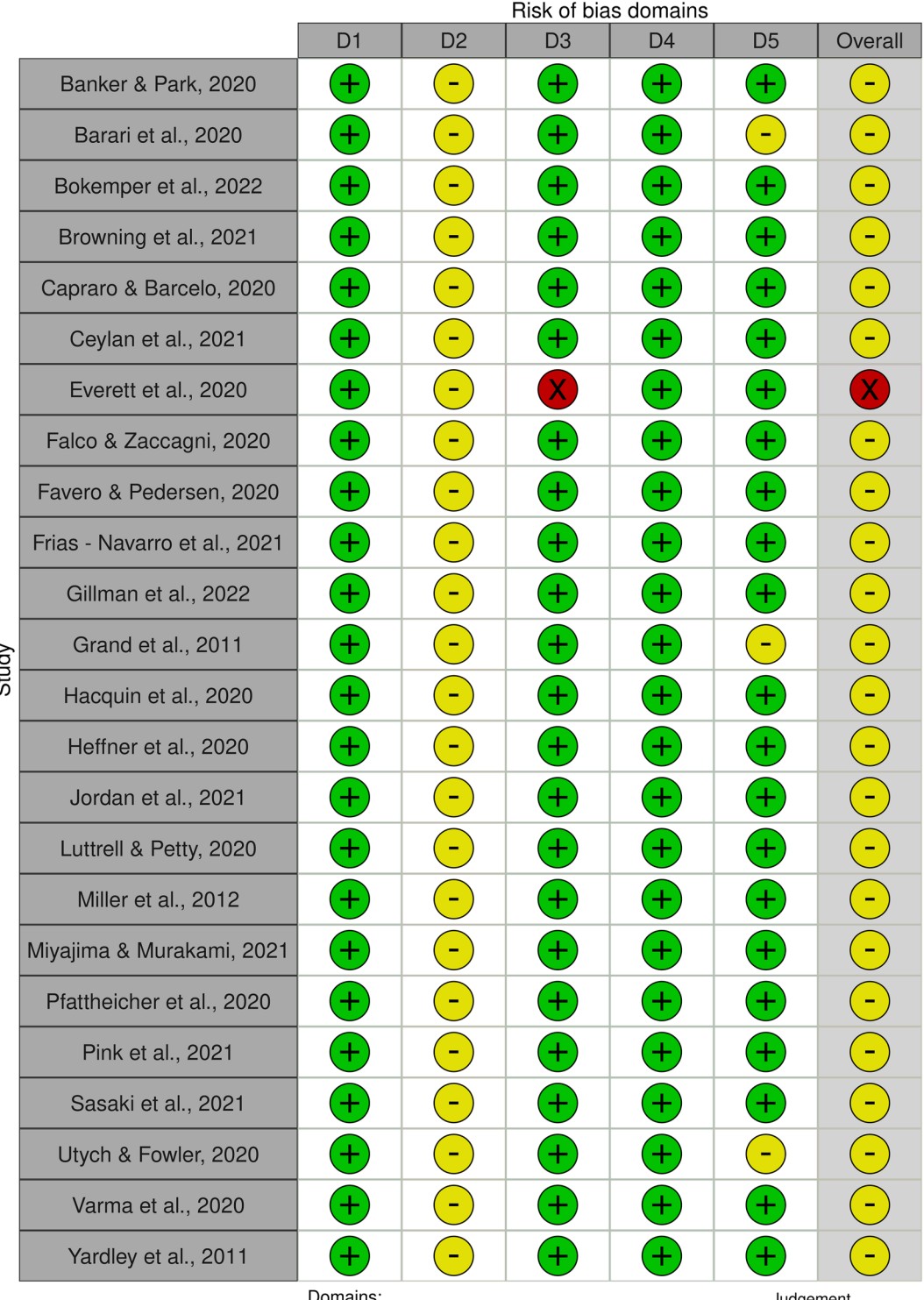

**Fig. 4 | Risk of bias assessment. Traffic-light plot of the domain-level judgements.**

in the population of interest. There was relatively low heterogeneity (SD = 0.07, 95% CrI 0.04 to 0.11), therefore, there was limited variability between studies in the analysis. The mean value of the total residual deviance (69.52) was similar to the number of data points (65) in the analysis, which indicates a reasonable fit (Table 1; Supplementary Fig. S6).

## Subgroup Network Meta-analyses by behavioural intention outcomes

There was no inconsistency identified between direct and indirect evidence for social distancing intentions (design-by-treatment interaction model: $\chi^2(6) = 3.704$, $p = 0.717$), mask wearing intentions (design-by-treatment interaction model: $\chi^2(4) = 4.585$, $p = 0.333$), diverse-behavioural

**Table 1 | Effect estimates on personal protective behavioural intentions**

| Comparison | Effect estimates: SMD | 95% CrI | Confidence rating |
|---|---|---|---|
| Prosocial message focused on public | 0.08 | 0.04, 0.12 | Low |
| Prosocial message focused on loved ones | 0.09 | 0.06, 0.14 | Low |
| Self-focused message | 0.08 | 0.03, 0.13 | Low |
| Model fit and heterogeneity | Between-study SD 0.07 | 95% CrI 0.04, 0.11 | |

Total residual deviance: Mean 69.52 from 65 datapoints; handwashing outcomes were not included due to inconsistency between direct and indirect evidence.

interventions (design-by-treatment interaction model: $\chi^2(3) = 0.140$, $p = 0.987$). However, there was evidence of inconsistency for handwashing intentions (design-by-treatment interaction model: $\chi^2(3) = 9.115$, $p = 0.028$). Local testing of evidence loops identified a statistically significant difference between direct and indirect evidence for the comparison between prosocial public messages and prosocial loved ones messages (d = 0.618, 95% CrI 0.04 to 1.19; $p = 0.03$).

### Effects on social distancing intentions
Thirteen RCTs were included in the random-effects NMA. There was a small increase in social distancing intentions for each intervention group compared to the control group. While the effects were small, the prosocial message focused on loved ones showed a slightly higher effect size (d = 0.10, 95% CrI 0.04 to 0.16, CINeMA: Moderate) than the prosocial message focused on the public. There was insufficient data to estimate an effect for the self-focused message (Supplementary Table S8; Supplementary Figs. S2, S7).

### Effects on mask wearing intentions
Four RCTs were included in the random-effects NMA. Each intervention group showed a small increase in mask-wearing intentions compared to the control group. Only prosocial messages focused on the public, however, demonstrated a slightly higher, statistically significant effect size (d = 0.16, 95% CrI 0.04 to 0.30, CINeMA: Moderate) (Supplementary Table S9; Supplementary Figs. S3, S8).

### Effects on handwashing intentions
Seven RCTs were included in the random-effects NMA. There was a small, however non-statistically significant increase in handwashing intentions for each intervention group compared to the control group, with a slightly higher effect to be observed for prosocial messages focused on loved ones (d = 0.20, 95% CrI −0.15 to 0.52, CINeMA: Low) (Supplementary Table S10; Supplementary Fig. S4; Supplementary Fig. S9).

### Effects on diverse-behavioural intentions
Nine RCTs were included in the random-effects NMA. There was a small increase in diverse-behavioural intentions for each intervention group compared to the control group. While the effects were small, the prosocial message focused on loved ones showed a slightly higher effect size (d = 0.17, 95% CrI 0.04 to 0.31, CINeMA: Low) than the prosocial message focused on the public. (Supplementary Table S11; Supplementary Figs. S5, S10) The estimated effect for the self-focused message was not statistically significant.

### Component network meta-analyses (CNMA) with Mindspace contextual influencers
Handwashing intention outcomes were excluded due to the inconsistency identified. The CNMA model had a reasonable fit (total residual deviance=57.96 from 59 data points) and low heterogeneity (SD = 0.05, 95% CrI 0.01 to 0.10). Although limited evidence, a small, non-statistically significant increase in personal protective behavioural intentions for interventions that incorporated the "salience" (d = 0.06, 95% CrI −0.04 to 0.16), "affect" (d = 0.06, 95% CrI −0.04 to 0.15) and "ego" (d = 0.05, 95% CrI −0.05 to 0.14) compared to interventions that did not include these contextual influencers was found (Table 2). To reduce potential heterogeneity, a

**Table 2 | CNMA effect estimates of MINDSPACE contextual influencers on personal protective behavioural intentions (prosocial messages)**

| Comparison | Effect estimates: SMD | 95% CrI |
|---|---|---|
| Messenger | −0.01 | −0.08, 0.05 |
| Norms | −0.04 | −0.26, 0.12 |
| Defaults | −0.04 | −0.24, 0.16 |
| Salience | 0.06 | −0.04, 0.16 |
| Affect | 0.06 | −0.04, 0.15 |
| Ego | 0.05 | −0.05, 0.14 |
| Model fit and heterogeneity | Between-study SD 0.05 | 95% CrI 0.01, 0.10 |

Total residual deviance: Mean 57.96 from 59 datapoints.

**Table 3 | CNMA effect estimates of BCTs on personal protective behavioural intentions (prosocial messages)**

| Comparison | Effect estimates: SMD | 95% CrI |
|---|---|---|
| 4.1. Instruction on how to perform the behaviour | −0.06 | −0.21, 0.02 |
| 5.1. Information about health consequences | 0.12 | 0.03, 0.28 |
| 5.2. Salience of consequences | 0.03 | −0.03, 0.07 |
| 5.3. Information about social and environmental consequences | 0.02 | −0.07, 0.10 |
| 6.1. Demonstration of the behaviour | 0.03 | −0.10, 0.26 |
| 6.2. Social comparison | −0.01 | −0.08, 0.07 |
| 7.1. Prompts/cues | 0.01 | −0.08, 0.11 |
| 9.1. Credible source | −0.01 | −0.06, 0.06 |
| 12.3. Avoidance/reducing exposure to cues for the behaviour | 0.19 | 0.06, 0.32 |
| Model fit and heterogeneity | Between-study SD 0.03 | 95% CrI 0.01, 0.07 |

Total residual deviance: Mean 56.53 from 59 datapoints.

separate CNMA for self-focused messages was conducted (Supplementary Table S12 and Supplementary Note 6).

### Component network meta-analyses with BCTs
Handwashing intention outcomes were excluded due to the inconsistency identified. The CNMA model had a reasonable fit (total residual deviance = 56.53 from 59 data points)) and low heterogeneity (SD = 0.03, 95% CrI 0.01 to 0.07). A small increase in personal protective behavioural intentions for interventions that incorporated the "information about health consequences" (d = 0.12; 95% CrI 0.03 to 0.28) and the "avoidance/reducing exposure to cues for the behaviour" (d = 0.19, 95% CrI 0.06, 0.32) compared to interventions that did not include these BCTs was found (Table 3). Separate CNMA for self-focused messages was conducted (Supplementary Table S13 and Supplementary Note 7).

https://doi.org/10.1038/s43856-025-01296-6                                                                                                    **Article**

## Discussion

Our systematic review supports the premise that public health messages focused on "protect-others" may help to improve respiratory infection control and may be more effective than self-interested messages. We identified 24 full-text articles, comprising 36 RCTs, conducted across 11 countries. Interventions were typically delivered online. An average of three-arm study designs with passive comparators were typically used. The overall quality of included studies was moderate, with only one study rated as 'high risk of bias'. In particular, 26 of the 36 RCTs found a significant effect of protect-others messages on outcomes such as social distancing intention, mask wearing intention, hand washing intention, diverse-behavioural intentions and actual behaviours.

The NMA results from 21 RCTs support our finding that prosocial messages have a small positive effect on personal protective behavioural intentions. Both types of prosocial messages showed a statistically significant effect on social distancing and diverse behavioural intentions, with messages highlighting loved ones showing slightly higher effectiveness, whereas only prosocial messages focusing on the public produced a statistically significant effect on mask-wearing intentions. None of the message types produced significant changes in handwashing intentions. Notably, while self-focused messages demonstrated a small but statistically significant effect on overall personal protective behavioural intentions in the combined analysis, they did not produce significant effects when outcomes, such as mask-wearing, handwashing, and diverse-behavioural intentions, were analysed separately. This suggests that the influence of self-focused messages may be more diffuse and less behaviour-specific. Thus, their standalone use may offer limited impact and could be deprioritised in future public health campaigns in favour of more targeted or emotionally resonant messaging strategies, such as those based on prosocial appeals. Numerous correlational studies indicate that intentions are associated with behaviour. In particular, current evidence suggests that intentions get translated into action approximately one-half of the time[37,80–82]. For example, Sheeran[83] conducted a meta-analysis that encompassed a total of 422 studies from ten previous meta-analyses. The outcome of this analysis revealed a substantial sample-weighted average correlation between intentions measured at one time-point and measures of behaviour taken at a subsequent time-point ($r+= 0.53$). It is essential to interpret these findings with caution, as reliance on intention measures without robust behavioural follow-up may lead to overly optimistic conclusions about message effectiveness. This highlights the need for interventions that go beyond motivation alone, by fostering enabling environments, incorporating reinforcement strategies, and addressing structural barriers to effectively bridge the intention–behaviour gap.

The systematic review also suggested some demographic characteristics to be predictors of protective behavioural intentions regarding respiratory infections. Women, older people, those having less secure employment, more religious people, liberals or politically left-leaning, and those who are in worse health conditions intend to enact protective behaviours. Moreover, the studies included in our analysis were conducted in high-income countries. Caution is therefore warranted when considering their applicability beyond these settings, highlighting the need for further investigation into the effectiveness of prosocial public health messaging interventions in middle- and low-income countries. Tailoring messages to local contexts and rigorously evaluating their impact in such settings should be an important focus for future research.

This review also examined which MINDSPACE contextual influencers and behaviour change techniques (BCTs) might underlie message effectiveness. Although many effective RCTs included contextual influencers such as "salience," "affect," and "ego" according to our narrative synthesis, the CNMA results did not support these findings. None of the contextual influencers were associated with statistically significant effects, although these three showed positive trends. Existing literature suggests that messages which increase the salience of risk can positively influence behaviours such as social distancing and mask-wearing[24]. Moreover, people are more likely to notice and respond to stimuli that are novel (e.g., messages in flashing lights), accessible, simple (e.g., a snappy slogan), and relevant (e.g., introduced at key moments such as the onset of a disease outbreak)[84]. For example, digital platforms like Instagram Stories have been shown to effectively disseminate public health messages, engaging diverse audiences and enhancing message memorability[85]. In addition, *ego* influences behaviour because individuals are motivated to act in ways that reinforce a positive and consistent self-concept (e.g., seeing themselves as responsible or protective of others). Research shows that appeals to self-image and the desire for social approval can effectively motivate action[84,86].

*Affect* strongly influences decision-making by bypassing deliberative processes and triggering rapid, intuitive responses[62,69]. Emotional messaging, delivered through images, metaphors, or relatable framing, can increase both engagement and retention[84,87]. For example, infectious-disease communications that evoke empathy for vulnerable groups have been shown to encourage protective intentions such as self-isolation and social distancing[72].

Similarly, although the narrative synthesis identified six BCTs (*instruction on how to perform a behaviour, information about health consequences, salience of consequences, information about social and environmental consequences, credible source,* and *avoidance/reducing exposure to cues for the behaviour*) as potentially important predictors of outcomes, the CNMA provided only partial support for these findings. Specifically, interventions incorporating "information about health consequences" and "avoidance/reducing exposure to cues" were associated with small but statistically significant increases in personal protective behavioural intentions. Theories suggest that associative (such as antecedents), reflective motivational (such as natural consequences and comparison of outcomes) and self-regulatory processes (such as shaping knowledge) BCTs are relevant to successful behaviour change[88]. In particular, motivation BCTs support behaviour change by strengthening individuals' intentions and making beliefs or feelings about the behaviour more favourable. Self-regulatory BCTs help individuals manage their thoughts, emotions, and behaviours to achieve goals, enabling them to act on their intentions within changing environments. Associative BCTs, by contrast, can trigger behaviours automatically, without conscious motivation or self-regulation, either aligning with existing goals or operating independently of them[89].

Investigating the MoAs through which interventions have their effects on behaviour is key for optimising their outcomes. The most frequently applied MoA through which the BCTs affected population behaviour was "behaviour intention". Identifying MoAs is crucial to understanding how the interventions change population behaviour.

However, while these elements may enhance the appeal and engagement of public health messages, the current evidence base does not allow for firm conclusions about their effectiveness. Their inclusion should therefore be regarded as exploratory, and further research is needed to determine whether, and under what conditions, these contextual features enhance behavioural impact.

In addition, for the elements that did not demonstrate evidence of effectiveness, this should not be taken to mean they are ineffective in all contexts. Instead, the findings suggest that in the reviewed studies, these components may not have been implemented in ways that were sufficiently strong, salient, or contextually tailored to produce meaningful behavioural change. Their limited impact may reflect issues of message design, delivery context, or audience relevance. Future research should investigate whether these components can be rendered more effective through co-design with target populations, tailored framing, repeated exposure, or integration into multi-component interventions that address multiple motivational pathways.

Our electronic search included four databases of published papers, several databases of "grey" literature. Our hand search involved the use of search engines such as Google Scholar, relevant journals and backward and forward citation searching. We re-ran the same search after a year to include new studies of interest.

Data extraction was initially performed by one reviewer and independently verified by a second. Outcome data were extracted independently by two reviewers, with any discrepancies resolved through discussion with a

third reviewer, thereby minimising errors and reducing the risk of bias. All abstracts, regardless of the language of the original studies, were initially screened for eligibility, as all were written in English. However, none of the studies in languages other than English met the inclusion criteria during the screening process.

This review used an empirically developed MINDSPACE checklist and a taxonomy of intervention techniques to code contextual influencers and BCTs, respectively, present in prosocial public health messaging interventions targeting personal protective behavioural intentions and actual behaviours. The MoAs Ontology was also used to identify the MoAs through which the BCTs can affect behaviour. Two coders independently coded all studies for contextual influencers and BCTs, and three coders independently coded the MoAs, which represents a significant strength of the review.

However, it is possible that several MINDSPACE contextual influencers, BCTs, and MoAs were not captured due to inadequate descriptions of interventions. Our analyses were based on the information presented in the published papers and any supplementary materials, though attempts were made to contact study authors for more detailed descriptions. Furthermore, since most interventions targeted multiple behaviours without clearly distinguishing which contextual influencers and BCTs were intended to affect specific behaviours, it was not possible to assess the extent of overlap between MINDSPACE elements and BCTs in this review.

Several additional limitations should be noted. The majority of included studies were one-off interventions, with only two studies conducting longer-term follow-ups. As a result, we were unable to assess whether prosocial public health messaging interventions have effects that extend beyond the immediate post-intervention period. Although the systematic review indicated that certain demographic characteristics may predict protective behavioural intentions in the context of respiratory infections, future research is needed to explore how the "protect others" principle influences behavioural responses across more diverse populations.

As expected, given the diversity of included studies, there was considerable methodological and statistical heterogeneity. Studies employed inconsistent methods of outcome aggregation, used varying metrics, and assessed outcomes at different time points, which limited direct comparability. To address these challenges, we applied the GRADE and CINeMA frameworks to systematically assess—and, where appropriate, downgrade—the certainty of evidence based on factors such as within-study bias, publication bias, indirectness, imprecision, heterogeneity, and incoherence. The use of these tools enhanced the methodological rigour of our review and strengthened the reliability of the NMAs.

Not all eligible studies could be included in the NMAs due to poorly reported results. The small number of studies with adequate data also prevented subgroup analyses (e.g., by population type or outcome assessment), which would have provided a deeper understanding of intervention effectiveness.

In addition, only six studies were designed to examine actual behaviours as their primary outcomes[16,59,61,73,75,76]. The majority measured intentions to engage in preventive procedures, limiting our ability to assess whether intentions translated into action (behaviour – intention gap[37,73,90,91]). As a result, a limitation of the network meta-analysis (NMA) presented here is that only studies reporting behavioural intentions could be included, as none provided actual behavioural outcomes in sufficient detail. Although intention is a well-established precursor to behaviour, empirical evidence indicates that intentions translate into action only about half of the time Falco and Zaccagni[61], suggesting that a substantial proportion of individuals may not follow through on their intended behaviours. This limitation should be considered when interpreting the effectiveness of prosocial messaging interventions, as reliance on intention-only data may overestimate their real-world behavioural impact. To address this, a narrative synthesis was conducted for each of the personal protective behavioural intentions and actual behaviours. Narrative synthesis goes beyond the act of simply describing the main features of included studies and enables summaries of knowledge related to a specific review question[92]. However, as noted, gaps between intention and action continue to be a challenge. Future work should prioritise interventions explicitly designed to evaluate behavioural outcomes in order to more accurately assess the impact of prosocial messaging strategies.

Furthermore, most of the included studies were conducted during the COVID-19 pandemic (2020–2022), a period marked by unprecedented uncertainty, rapidly changing policies, and heightened public awareness. As a result, behaviours and messaging observed during this time may not fully reflect more stable, post-acute contexts, which could limit the generalizability of certain findings.

Finally, as all studies included in our analysis were conducted in high-income countries, caution is warranted when generalising the findings to other contexts. Future research should prioritise the evaluation of prosocial messaging interventions in middle- and low-income settings, where cultural, social, and structural factors may influence behavioural responses differently.

Our findings hold meaningful implications for public policymakers and communication experts, particularly regarding how best to communicate complex, uncertain, and sometimes conflicting health information. Strengthening the design of public health messages is vital for improving respiratory infection control and fostering protective behaviours. Evidence from this review suggests that framing messages around social responsibility, especially by highlighting the impact on loved ones (friends, family, and vulnerable community members), can be especially persuasive.

Effective risk communication involves raising public awareness not only about personal consequences but also the broader implications for others' health. Tailoring prosocial messages requires a nuanced understanding of individual differences in perceived susceptibility, emotional response, and message preference. Framing interventions to help people avoid or reduce exposure to behavioural cues have shown promise in supporting risk-reducing behaviours (Table 4).

## Table 4 | Policy Recommendations

| Recommendation | Description/rationale | Examples of Interventions and Strategies Based on Findings |
|---|---|---|
| Emphasise protection of loved ones | Messages referencing family or close relationships evoke stronger emotional engagement and prosocial motivation. | Use messaging like "Protect your family and friends," or visual narratives showing vulnerable loved ones impacted by risky behaviours. |
| Tailor content to audience segments | Demographic factors (e.g., age, gender, political orientation, health status) influence responsiveness to messaging. | Design for specific groups—empathetic appeals for older adults, community-oriented framing for conservatives, messages that address employment precarity. |
| Prioritise effective BCTs | Behaviour Change Techniques like "Information about health consequences" and "Avoidance/reducing exposure to cues" show modest impact on intentions. | Share clear health warnings, remove behavioural prompts (e.g., signage), and use credible messengers like healthcare professionals or trusted community leaders. |
| Measure behaviour —not just intention | Behavioural intentions only translate into action about half the time; relying solely on intention may overestimate impact. | Use behavioural outcomes such as observation, compliance follow-ups, or biometric proxies; triangulate with intention data to validate real-world effectiveness. |

## Data availability

This systematic review and meta-analysis is based on data extracted from publicly available studies. The authors declare that the data supporting the findings of this study are available within the paper, its supplementary information file and supplementary data files. The characteristics of all included studies are provided in Supplementary Data 2 and 3. The source data for Fig. 1 (PRISMA flow chart) can be found in the Supplementary Information. The source data for Figs. 2 and 3 (ERs of MINDSPACE contextual influencers and behaviour change techniques, respectively) can be found in Supplementary Data 5. The source data for Fig. 4 (risk of bias) can be found in Supplementary Data 4. The source data for Table 1 (network meta-analysis) can be found in Supplementary Data 6. The source data for Tables 2 and 3 (CNMA of MINDSPACE contextual influencers and behaviour change techniques, respectively) can be found in Supplementary Data 6.

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

## Acknowledgements

The authors thank Fiona Beyer for her expertise in developing the search strategy, as well as Paulina Schenk, for her contribution to the identification of mechanisms of action. The authors gratefully acknowledge Louise Tanner and Ryan Kenny for their time and consultation regarding the meta-analysis. This study/project is funded by the National Institute for Health Research (NIHR) [Policy Research Unit in Behavioural Science (project reference PR-PRU-1217-20501)]. The views expressed are those of the author(s) and not necessarily those of the NIHR or the Department of Health and Social Care.

## Author contributions

A.G. is the corresponding author and the primary author of the study, conceived the study, contributed to the development of the search strategy, developed the inclusion and exclusion criteria and data extraction criteria, was involved in the conceptualization of the research questions and conducted the systematic review (behavioural coding and analyses of the results) and meta-analysis. V.A. contributed to the development of the search strategy, was involved in the behavioural coding of the papers and in the meta-analyses, and provided written feedback on the manuscript. N.M. contributed to the Network Meta-analysis and provided written feedback on the manuscript. C.B. conceived the study, contributed to the development of the selection criteria and data extraction criteria, was involved in the conceptualization of the research questions and provided written feedback on the manuscript. S.M. conceived the study and provided written feedback on the manuscript. M.K. contributed to the development of the selection criteria and data extraction criteria and provided written feedback on the manuscript. I.V. conceived the study, contributed to the development of the selection criteria and data extraction criteria, was involved in the conceptualization of the research question and revised the manuscript critically and contributed to it intellectually. All authors have read and approved the final version of the manuscript.

## Competing interests

The authors declare no competing interests.

## Additional information

[1]NIHR Policy Research Unit in Behavioural Science – Behavioural Science Group, Warwick Business School, University of Warwick, Coventry, UK. [2]NIHR Policy Research Unit in Behavioural Science – Health Psychology Research Group, Department of Clinical, Education and Health Psychology, University College London, London, UK. [3]Population Health Sciences Institute, Newcastle University, Newcastle, UK. [4]NIHR Policy Research Unit Behavioural Science - Department of Public Health, Environments and Society, London School of Hygiene and Tropical Medicine, London, UK. [5]Department of Public Health and Primary Care, University of Cambridge, Cambridge, UK. [6]NIHR Policy Research Unit Behavioural Science – Population Health Sciences Institute, Faculty of Medical Sciences, Newcastle University, Newcastle, UK. [7]Centre for Behavioural and Implementation Science Interventions (BISI), Yong Loo Lin School of Medicine, National University of Singapore, Singapore, Singapore. ✉e-mail: aikaterini.grimani@wbs.ac.uk

