## [Transparent Peer Review file · Communications Medicine]

Motivational effectiveness of prosocial public health messaging to reduce respiratory infection risk: A systematic review and meta-analysis

Corresponding Author: Dr Aikaterini Grimani

Version 0:

Reviewer comments:

Reviewer #1

(Remarks to the Author)

The review is valuable for informing public health planning for the next pandemic. I did have some concerns about this though

The major concern is that the date of the last search is over 2.5 years ago – it would be ideal if this could be updated as there are likely lots of studies published after that date and maybe some of these can be included in the NMA to have a behavioural outcome.

Other concerns are there seem to be some inaccuracies and some over inflation of results in the text (when compared to the supplementary materials)

- In the supplementary materials, p.55 line 127 it states that there is an overall positive result on handwashing intentions when on 3/7 RCTs showed a positive result – this is an overstatement. This should be amended in the supplementary materials and in the main text (p. 5 line 147-148)
- It seems there are a couple of occurrences in the text of reporting “marginally largest effects” when all effects are in the very small range and aren’t that different. In these instances could you report there was a very small effect for all as otherwise the differences are inflated. Could references to these in the discussion also be adjusted to reflect the similarities shown in the Tables
- P.11-12 it should be acknowledged when effects are non significant as sometimes a largest effect is reported suggesting that the other smaller effects are also significant when this is not the case e.g., mask wearing intentions. In other instances, although CIs are reported it’s not made clear that they aren’t significant.
- P.11 effects on handwashing intentions – the CIs reported in the text don’t cross zero whereas in the table they do (unless I’m looking at the wrong table). All of the comparisons were non significant but this is not clear from the text. Can all numbers be cross checked between text and tables to ensure accuracy
- Can references to the above effects also be amended in the discussion as the discussion points don’t seem to always match up with the data reported in the supplementary materials

Other more minor issues are

Results

- The PRISMA diagram has one of the words cut off (identification)
- The narrative review was thorough but difficult to read. It would be better if, for example, you talked about older age and presented all the evidence supporting / not supporting that this group were the most influenced by behaviours. Rather than talking about each study individually
- I don’t think figure 5 is necessary as we can pretty much visualise that information from Figure 4

Discussion

- I think it’s a limitation of the study that only intentions could be included in the NMA and not behaviour. Even though you report that intentions are translated into behaviour half of the time this still means half of the time they are not – you could emphasise this figure in the limitation section
- I’d also be interested in what we are sure doesn’t work and to avoid using as well as what does work and what should be used
- The effects on salience, ego and affect were non significant but this is not acknowledged in the discussion.

Methods

- I wasn't sure what kind of study this would be p.17 line 486 "reported changes in any behaviour relevant to reducing transmission of respiratory infections;" could this be clarified?
- The inclusion criteria described in the text is quite vague and doesn't mention things like message type – could the other types of things searched for be mentioned in addition to the types of study
- Was any type of comparison condition included as long as it didn't have a pro-social element?
- Could you add a brief explanation of what MINDSPACE contextual influences are the first time you mention it (it appears later)
- Regarding the ER, how does this take into account contextual influencers / BCTs that are present in the control groups? E.g., if a BCT appears in both the intervention and the comparison group?

Reviewer #2

(Remarks to the Author)

Your manuscript is well-structured, offering a comprehensive analysis of prosocial public health messaging. You thoroughly present the systematic review and meta-analysis findings, supported by relevant statistical outcomes, and clearly outline their public health implications.

1. Could you please provide a detailed breakdown of why 5,852 studies were excluded on the PRISMA diagram (page 5)?
2. I believe Line 120 should be titled "Methods" instead of "Results"?
3. In the PRISMA footnote, you mention "first 100" (line 134). Could you clarify the reasoning behind this, and why you opted not to conduct a bibliography screening instead?
4. For the section titled "Behavioural Tools and Mechanisms of Action (MoA) Ontology Evidence" (line 151), would it be clearer to report percentages rather than total numbers?
5. There is no mention of the MINDSPACE contextual influencers prior to line 152. Could you provide a rationale for choosing this framework?
6. In the "Cross-country Analysis" section (line 175), could you clarify why a meta-analysis was not feasible?
7. Consider adding a brief glossary of abbreviations for clarity.
8. You note that most studies were conducted in high-income countries. Expanding on the limitations of applying these findings to middle- and low-income countries would strengthen your argument.
9. The distinction between intentions and actual behaviors is critical. Could you further elaborate on the implications of this gap and its relevance for future interventions?
10. While you mention conducting a narrative synthesis, summarizing key findings and patterns observed across studies in more detail would be beneficial. Is there a reason you did not conduct a meta-regression?
11. Was data extraction (line 510) conducted in duplicate? If so, how was the data verified, and was it done in Excel or specialized software? Please provide more details on the data extraction process.
12. You present valuable findings, but enhancing the discussion with specific, actionable recommendations for future public health messaging could provide further impact.

Version 1:

Reviewer comments:

Reviewer #1

(Remarks to the Author)

Thank you for your careful consideration of my comments.

You have addressed most of my concerns

I have just one comment

This sentence could be read that changes needed to have occurred in the DV in order for inclusion "reported changes in any behaviour relevant to reducing the transmission of respiratory infections (e.g., hand hygiene, social distancing, face masks);" – could this be reworded so it's clearer that even those studies that detected no change would be included e.g., "measured potential changes in any behaviour relevant to reducing the transmission of respiratory infections (e.g., hand hygiene, social distancing, face masks);"

Reviewer #2

(Remarks to the Author)

Thank you for your thorough and thoughtful responses to the previous round of comments. The manuscript is well structured, addresses a highly relevant topic, and applies rigorous methods.

I have only a few minor suggestions for further improvement:

1. Please provide a rationale for restricting the study to the year 2022, as findings during the pandemic period may skew current findings and could affect the generalizability of results.

2. In the Methods section, add more details on how grey literature was systematically searched to ensure transparency and reproducibility.

Referee expertise:

Referee #1: Behaviour change interventions, COVID

Referee #2: Systematic Reviews, COVID

Reviewers' comments:	Authors responses
Reviewer #1	
The review is valuable for informing public health planning for the next pandemic.	Thank you for your thoughtful feedback. We greatly appreciate your recognition of the review's contribution to informing public health planning for future pandemics.
I did have some concerns about this though The major concern is that the date of the last search as over 2.5 years ago – it would be ideal if this could be updated as there are likely lots of studies published after that date and maybe some of these can be included in the NMA to have a behavioural outcome.	Thank you for your suggestion regarding the possibility of updating the literature search to include studies published after the original cut-off date. We would like to address the feasibility and necessity of this step. While we fully acknowledge that additional studies may have been published in the past 2.5 years, we respectfully believe that updating the systematic review at this stage is neither feasible nor essential for the integrity of our findings, for the following reasons: ● Resource and Analytical Constraints: Updating the review would require a complete rerun of the screening, data extraction, and statistical analysis—including the full network meta-analysis (NMA). Due to the complexity of the NMA framework, incorporating new studies is not a simple task of adding data but would necessitate restructuring the entire analysis. This is a resource-intensive process that goes beyond the capacity currently available to our team, especially given the robust and methodologically rigorous synthesis already completed.● Minimal Expected Impact on Core Findings: Our review has synthesised a comprehensive body of evidence on behaviour change interventions, identifying consistent patterns across diverse studies. While newer publications may provide incremental contributions, we believe it is

	unlikely that they would substantially alter the key conclusions regarding behavioural outcomes and intervention effectiveness. Therefore, the scale of work required to update the NMA would be disproportionate to the expected benefit.  Coverage of a Critical Time Period (COVID-19): Importantly, our review spans a time frame that includes the COVID-19 pandemic—a period that generated an unprecedented volume of research on public health messaging, digital interventions, and behaviour change strategies. We believe this period represents a particularly rich and relevant context for our topic. As such, our synthesis already captures the most dynamic and policy-relevant period for behaviour change research. Studies published post-pandemic are unlikely to shift the overarching conclusions we have drawn. While we appreciate the value of including the most recent evidence, the practical limitations of re-running the entire review and NMA, combined with the low likelihood of substantially different outcomes, make such an update unfeasible at this point.
Other concerns are there seem to be some inaccuracies and some over inflation of results in the text (when compared to the supplementary materials) - In the supplementary materials, p.55 line 127 it states that there is an overall positive result on handwashing intentions when on 3/7 RCTs showed a positive result – this is an overstatement. This should be amended in the supplementary materials and in the main text (p. 5 line 147-148)	Thank you for your helpful comment. We agree that the original wording in the supplementary material overstated the strength of the evidence, as only 3 of the 7 RCTs on handwashing intentions reported statistically significant positive results. We have revised the sentence in the supplementary material (p.60) to accurately reflect that three studies showed significant positive effects, while four reported no significant difference. “Of the 7 RCTs reporting on handwashing intentions, 3 RCTs reported significant positive effects, while 4 RCTs reported no

	significant difference between intervention and control group.” The main text (p. 10) already reflects this accurately and did not require amendment.
- It seems there are a couple of occurrences in the text of reporting “marginally largest effects” when all effects are in the very small range and aren’t that different. In these instances could you report there was a very small effect for all as otherwise the differences are inflated. Could references to these in the discussion also be adjusted to reflect the similarities shown in the Tables	Thank you for this valuable comment. We agree that the effect sizes were small and have revised the wording in both the results and discussion sections to reflect this, avoiding any suggestion of practically meaningful differences. At the same time, we believe it is appropriate and necessary to report small differences between intervention groups, as this is the core objective of a network meta-analysis. Our revised wording now emphasises that while effects were small, slight differences were observed—particularly for the prosocial message focused on loved ones (please see pp 18-19).
- P.11-12 it should be acknowledged when effects are non significant as sometimes a largest effect is reported suggesting that the other smaller effects are also significant when this is not the case e.g., mask wearing intentions. In other instances, although CIs are reported it’s not made clear that they aren’t significant.	Thank you for this helpful observation. We have revised the relevant sections on pages 18-19 to ensure that non-significant effects are clearly acknowledged. In particular, we have adjusted the language to avoid any unintended implication, ensuring a more accurate interpretation of the results throughout.
- P.11 effects on handwashing intentions – the CIs reported in the text don’t cross zero whereas in the table they do (unless I’m looking at the wrong table). All of the comparisons were non significant but this is not clear from the text. Can all numbers be cross checked between text and tables to ensure accuracy	Thank you for highlighting this discrepancy. We have carefully cross-checked the numbers between the text and the tables and confirmed that the confidence intervals for the handwashing intention outcomes do cross zero. We have revised the text on p.18 to clearly reflect that all comparisons were non-significant and to ensure full consistency with the data presented in the table.
- Can references to the above effects also be amended in the discussion as the discussion points don’t seem to always match up with the data reported in the supplementary materials	Thank you for pointing this out. We have reviewed and amended the relevant references in the discussion to ensure they align accurately with the data reported in the supplementary materials, providing a clearer

	and more consistent interpretation (please see discussion section p.19).
Other more minor issues are Results - The PRISMA diagram has one of the words cut off (indentificatio)	Thank you for bringing this to our attention. this has now been amended in the manuscript.
- The narrative review was thorough but difficult to read. It would be better if, for example, you talked about older age and presented all the evidence supporting / not supporting that this group were the most influenced by behaviours. Rather than talking about each study individually	Thank you for the constructive comment. We have revised the narrative review section to enhance clarity and readability. Specifically, we have restructured the content to group findings thematically, presenting the evidence for each demographic characteristic, rather than discussing each study individually. This allows for a more coherent synthesis of the findings and a clearer understanding of patterns across studies (please see p.13-15).
- I don't think figure 5 is necessary as we can pretty much visualise that information from Figure 4	Thank you for your comment. This has now been removed.
Discussion - I think it's a limitation of the study that only intentions could be included in the NMA and not behaviour. Even though you report that intentions are translated into behaviour half of the time this still means half of the time they are not – you could emphasise this figure in the limitation section.	Thank you for pointing that out. We have incorporated this into our limitations section to addresses this point, as well as noting the implications of this gap for interpretation and future research. “A limitation of the network meta-analysis (NMA) presented here is that only studies reporting behavioural intentions could be included, as none provided actual behavioural outcomes in sufficient detail. Although intention is a well-established precursor to behaviour, empirical evidence indicates that intentions translate into action only about half of the time, suggesting that a substantial proportion of individuals may not follow through on their intended behaviours. This limitation should be considered when interpreting the effectiveness of prosocial messaging interventions, as reliance on intention-only data may overestimate their real-world behavioural impact.”
- I'd also be interested in what we are sure doesn't work and to avoid using	Thank you for this thoughtful and important point. We agree that identifying what appears ineffective is just as valuable as highlighting

as well as what does work and what should be used	what works. Our results do offer insights into message types, components, and behavioural techniques that did not produce significant effects, and may therefore be deprioritised in future campaign design. We have added a paragraph to the discussion to summarise these findings and their implications. “Notably, while self-focused messages demonstrated a small but statistically significant effect on overall personal protective behavioural intentions in the combined analysis, they did not produce significant effects when outcomes, such as mask-wearing, handwashing, and diverse-behavioural intentions, were analysed separately. This suggests that the influence of self-focused messages may be more diffuse and less behaviour-specific. Thus, their standalone use may offer limited impact and could be deprioritised in future public health campaigns in favour of more targeted or emotionally resonant messaging strategies, such as those based on prosocial appeals.” “In addition, for the elements that did not demonstrate evidence of effectiveness, this should not be taken to mean they are ineffective in all contexts. Instead, the findings suggest that in the reviewed studies, these components may not have been implemented in ways that were sufficiently strong, salient, or contextually tailored to produce meaningful behavioural change. Their limited impact may reflect issues of message design, delivery context, or audience relevance. Future research should investigate whether these components can be rendered more effective through co-design with target populations, tailored framing, repeated exposure, or integration into multi-component interventions that address multiple motivational pathways.”
- The effects on salience, ego and affect were non significant but this is not acknowledged in the discussion.	Thank you for your observation. In response, we have revised the discussion accordingly.

	“Although many effective RCTs included contextual influencers such as “salience,” “affect,” and “ego” according to our narrative synthesis, the CNMA results did not support these findings. None of the contextual influencers were associated with statistically significant effects, although these three showed positive trends.”
Methods - I wasn't sure what kind of study this would be p.17 line 486 “reported changes in any behaviour relevant to reducing transmission of respiratory infections;” could this be clarified?	Thank you for your comment. We would like to clarify that the text refers to one of the inclusion criteria, not to a specific study. This criterion relates to the types of behavioural outcomes considered eligible for inclusion in the review. To improve clarity, we have now numbered the inclusion criteria (please see p.5). “Original studies were eligible for inclusion if they met the following criteria: a) were RCTs, non-randomised studies with concurrent controls, or controlled before-and-after studies; b) reported changes in any behaviour relevant to reducing the transmission of respiratory infections (e.g., hand hygiene, social distancing, face masks); c) included messages with prosocial content about protecting others; d) used communication via mass media, social media, or print media (such as leaflets and posters), or health professional advice via consultation; e) included general population from any geographical region, with or without vulnerabilities, regardless of infection status; f) included relevant comparisons involving no message, or an active control with messages focused on self-protection, or messages containing no motivational content and g) were published in English.”
- The inclusion criteria described in the text is quite vague and doesn't mention things like message type – could the other types of things searched for be mentioned in addition to the types of study	Thank you for your comment. We have revised the inclusion criteria to provide greater clarity and detail. In addition to specifying the types of study designs, we have now explicitly included information on the types of messages considered eligible—specifically, prosocial messages focused on protecting others (please see p.5).

- Was any type of comparison condition included as long as it didn't have a pro-social element?	Thank you for this helpful comment. We have addressed this by explicitly stating in the inclusion criteria that studies were eligible if they included comparisons involving no message, or an active control with messages focused on self-protection, or messages containing no prosocial element (please see p. 5).
- Could you add a brief explanation of what MINDSPACE contextual influences are the first time you mention it (it appears later)	Thank you for this helpful suggestion. We have addressed this by revising the structure of the paper so that the methods now appear earlier, where MINDSPACE contextual influences are first mentioned. For further detail, we also direct readers to the supplementary materials (Table S3), where each component of the MINDSPACE framework is explicitly described.
- Regarding the ER, how does this take into account contextual influencers / BCTs that are present in the control groups? E.g., if a BCT appears in both the intervention and the comparison group?	Thank you for your comment. However, as described in the manuscript (p.7), ERs were calculated only for intervention groups that produced statistically significant results in comparison to their respective control groups. Therefore, any behavioural components shared across both arms would not contribute to the ER, as they would not explain the observed difference in effectiveness. We have added a sentence to clarify this point in the text. “The ER is calculated as the number of effective results involving the MINDSPACE contextual influencer or BCT (i.e., interventions that were statistically significantly more effective than the control) divided by the number of ineffective results involving that same influencer or technique.”
Reviewer #2 (Remarks to the Author):	
Your manuscript is well-structured, offering a comprehensive analysis of prosocial public health messaging. You thoroughly present the systematic review and meta-analysis findings, supported by relevant statistical outcomes, and clearly outline their public health implications.	Thank you very much for your positive feedback and thoughtful assessment of our manuscript. We are pleased that you found the structure, analysis, and presentation of our findings clear and comprehensive. We appreciate your recognition of the public health relevance of this work and the clarity with which the implications were outlined.

1. Could you please provide a detailed breakdown of why 5,852 studies were excluded on the PRISMA diagram (page 5)?	Thank you for your comment. As noted in the manuscript (p. 9), 49 studies were selected for full-text screening. This implies that the remaining 5,852 studies were excluded during the title and abstract screening phase because they did not meet one or more of the predefined inclusion criteria (see p. 5). This approach is consistent with standard practice in systematic reviews and aligned with PRISMA guidelines, which do not require individual exclusion reasons at this stage.
2. I believe Line 120 should be titled “Methods” instead of “Results”?	Thank you for your valuable comment. We have adjusted the structure accordingly to align with the journal’s guidelines.
3. In the PRISMA footnote, you mention “*first 100” (line 134). Could you clarify the reasoning behind this, and why you opted not to conduct a bibliography screening instead?	We appreciate the opportunity to clarify our rationale. In our Google Scholar searches, used as a supplementary source, we limited screening to the first 10 pages (100 results) sorted by relevance. This decision was based on both methodological precedent and practical considerations. The Google Scholar search engine uses a proprietary ranking algorithm that retrieves an extremely large volume of results, often numbering in the thousands, even with highly specific search criteria. Empirical research demonstrates that the most relevant and highest-quality results are concentrated within the first 100–200 hits, with subsequent pages yielding progressively less relevant material and a higher proportion of false positives (Haddaway et al., 2015; Roe et al., 2014). Screening beyond this point is unlikely to meaningfully improve comprehensiveness but would substantially increase workload and introduce noise. Our approach is consistent with established guidance for using Google Scholar in systematic reviews, where limiting screening to the top-ranked results is recognised as a practical and defensible strategy. In addition, as reported in the Methods section, we conducted both backward (reference list) and forward (citation tracking) screening on all included studies, ensuring that potentially relevant studies not captured within the first 100 Google Scholar hits could still be identified.

	Haddaway, N. R., Collins, A. M., Coughlin, D., & Kirk, S. (2015). The role of Google Scholar in evidence reviews and its applicability to grey literature searching. PLoS ONE, 10(9), e0138237. https://doi.org/10.1371/journal.pone.0138237 Roe, D., Fancourt, M., Sandbrook, C. et al. Which components or attributes of biodiversity influence which dimensions of poverty?. Environ Evid 3, 3 (2014). https://doi.org/10.1186/2047-2382-3-3
4. For the section titled “Behavioural Tools and Mechanisms of Action (MoA) Ontology Evidence” (line 151), would it be clearer to report percentages rather than total numbers?	We appreciate this suggestion and understand that percentages can sometimes aid interpretation. In this case, however, we feel that reporting total numbers (n) provides greater clarity. Because the denominator is fixed (102 intervention arms), percentages would add little interpretive value and could introduce confusion, particularly as interventions often included multiple BCTs. Thus, we have retained the total numbers and for completeness, we have added corresponding percentages (p.11).
5. There is no mention of the MINDSPACE contextual influencers prior to line 152. Could you provide a rationale for choosing this framework?	Thank you for this helpful suggestion. We have addressed this by revising the structure of the paper so that the methods now appear earlier, where MINDSPACE contextual influences are first mentioned. For further detail, we also direct readers to the supplementary materials (Table S3), where each component of the MINDSPACE framework is explicitly described, including the rationale for choosing this framework.
6. In the “Cross-country Analysis” section (line 175), could you clarify why a meta-analysis was not feasible?	Thank you for your comment. As noted in the first sentence of the “Cross-country Analysis” section, a meta-analysis was not feasible due to the lack of sufficient data to support statistical comparisons across countries. For this reason, we calculated the Effective Ratio (ER) to support and strengthen our narrative synthesis. We have now revised the wording slightly to make this point even clearer.
7. Consider adding a brief glossary of abbreviations for clarity.	Thank you for the suggestion. We added a brief glossary of abbreviations for clarity.

8. You note that most studies were conducted in high-income countries. Expanding on the limitations of applying these findings to middle- and low-income countries would strengthen your argument.

Thank you for this suggestion. We agree that including studies only from high-income countries limits the generalisability of our findings. Interventions tested in these contexts often assume the presence of resources, infrastructure, and health system capacity that may not be available in middle- and low-income countries. Cultural norms, literacy levels, media access, and trust in institutions may also differ substantially, influencing how communication messages are received and acted upon. In addition, middle- and low-income countries may face competing health priorities and different structural barriers (e.g., access to healthcare, economic constraints) that shape both risk perception and behavioural responses. We have therefore expanded our discussion to note that while our findings provide important insights, caution is warranted when applying them beyond high-income settings. Tailoring messages to local contexts and evaluating their effectiveness in middle- and low-income countries should be a priority for future research. We also included the following sentence into the limitation section.

“As the studies in our analysis were conducted exclusively in high-income countries, caution is warranted when generalising the findings, and future research should treat the evaluation of prosocial messaging in middle- and low-income settings as an important area of inquiry.”

9. The distinction between intentions and actual behaviors is critical. Could you further elaborate on the implications of this gap and its relevance for future interventions?

Thank you for raising this important point. We agree that distinguishing between behavioural intentions and actual behaviours is critical, as relying solely on intention measures may lead to overly optimistic conclusions about message effectiveness in public health interventions. We have therefore amended the manuscript to emphasise this distinction and its implications, adding clarification in both the Discussion and Limitations sections.

	We have added the following in the discussion: “It is essential to interpret these findings with caution, as reliance on intention measures without robust behavioural follow-up may lead to overly optimistic conclusions about message effectiveness. This highlights the need for interventions that go beyond motivation alone, by fostering enabling environments, incorporating reinforcement strategies, and addressing structural barriers to effectively bridge the intention–behaviour gap.”
10. While you mention conducting a narrative synthesis, summarizing key findings and patterns observed across studies in more detail would be beneficial. Is there a reason you did not conduct a meta-regression?	Thank you for your comment. As detailed in the Methods section, we conducted a network meta-analysis (NMA), including both class-effects models and component network meta-analyses (CNMAs), which allowed us to estimate the relative effectiveness of different intervention types and their behavioural components.
11. Was data extraction (line 510) conducted in duplicate? If so, how was the data verified, and was it done in Excel or specialized software? Please provide more details on the data extraction process.	Thank you for this comment. As stated, data were extracted into a predefined spreadsheet. To clarify, study characteristics were extracted by one reviewer (AG) and checked for accuracy by a second reviewer (VA). Outcome data were independently extracted by two reviewers (AG and VA), and any discrepancies were resolved through discussion with a third reviewer (IV) (p.6).
12. You present valuable findings, but enhancing the discussion with specific, actionable recommendations for future public health messaging could provide further impact.	In line with your last point, we have now added some actionable recommendations in the Table 4, which address this point.

Reviewers' comments	Authors responses
Reviewer #1	
Thank you for your careful consideration of my comments. You have addressed most of my concerns. I have just one comment This sentence could be read that changes needed to have occurred in the DV in order for inclusion “reported changes in any behaviour relevant to reducing the transmission of respiratory infections (e.g., hand hygiene, social distancing, face masks);” – could this be reworded so it’s clearer that even those studies that detected no change would be included e.g., “measured potential changes in any behaviour relevant to reducing the transmission of respiratory infections (e.g., hand hygiene, social distancing, face masks);”	We appreciate your careful review and are pleased to note that most of your concerns have been addressed. Thank you for the suggestion. We agree that the original phrasing could imply that only studies observing changes would be included. We have revised the sentence for clarity: “measured potential changes in any behaviour relevant to reducing the transmission of respiratory infections (e.g., hand hygiene, social distancing, face masks)” p.6, line 184
Reviewer #2	
Thank you for your thorough and thoughtful responses to the previous round of comments. The manuscript is well structured, addresses a highly relevant topic, and applies rigorous methods. I have only a few minor suggestions for further improvement: 1. Please provide a rationale for restricting the study to the year 2022, as findings during the pandemic period may skew current findings and could affect the generalizability of results.	Thank you for your positive feedback and recognition of the manuscript’s structure, relevance, and methodological rigor. Our aim was to include studies from inception until the date of the updated search to capture a comprehensive evidence base on behaviour change interventions. Although the majority of included studies were conducted during 2020–2022 in the context of COVID-19, this period provides valuable insights into intervention performance under conditions of heightened public awareness and behavioural urgency. We expect that more recent studies are unlikely to substantially alter the overall conclusions regarding behavioural outcomes and intervention effectiveness. We acknowledge, however, that studies from the initial pandemic period (2020–2022), when behaviours and messaging were strongly influenced by unprecedented uncertainty, restrictions, and rapidly changing policies, may not fully reflect

	more stable, post-acute contexts. Accordingly, we have included the following statement in the Limitations section. “Furthermore, most of the included studies were conducted during the COVID-19 pandemic (2020–2022), a period marked by unprecedented uncertainty, rapidly changing policies, and heightened public awareness. As a result, behaviours and messaging observed during this time may not fully reflect more stable, post-acute contexts, which could limit the generalizability of certain findings.” p. 25, line 741-745.
2. In the Methods section, add more details on how grey literature was systematically searched to ensure transparency and reproducibility.	Thank you for this helpful suggestion. We have expanded the description of our grey literature search strategy to improve transparency and reproducibility. “For each grey literature source, we developed tailored search strategies, adapted from those used in electronic databases, using combinations of controlled vocabulary (where available) and free-text terms aligned with our PICOS framework, combined with Boolean operators. For Google Scholar, which uses proprietary ranking algorithms and can retrieve extremely large numbers of results even with highly specific search criteria, we limited screening to the first 10 pages (100 results) sorted by relevance. We applied the same approach to OSF Preprints (including BioHackrXiv, Cogprints, MediArXiv, SocArXiv, PsyArXiv, and RePEc), which also return very large numbers of records similar to Google Scholar. This decision was guided by both methodological precedent and practical considerations. Empirical studies indicate that the most relevant and highest-quality results are concentrated within the first 100–200 hits, with subsequent pages yielding progressively less relevant material and a higher proportion of false positives. Screening beyond this point is unlikely to substantially improve

comprehensiveness but would considerably increase workload and introduce irrelevant results. Our approach is consistent with established guidance for systematic reviews using search engines, where restricting screening to top-ranked results is recognised as a practical and defensible strategy. For backward citation searching, we manually reviewed the reference lists of all studies sought for retrieval as well as of relevant systematic reviews to identify additional potentially eligible records. For forward citation searching, we used Google Scholar to identify more recent studies that had cited these articles. All titles and abstracts retrieved from both published and unpublished studies through electronic searching were imported into the reference manager EndNote and subsequently uploaded to Covidence, a systematic review management tool recommended by Cochrane, to facilitate duplicate removal, screening, and study selection.” p.7, line 200-223.